# Gut Dysbiosis during COVID-19 and Potential Effect of Probiotics

**DOI:** 10.3390/microorganisms9081605

**Published:** 2021-07-28

**Authors:** Yuan-Pin Hung, Ching-Chi Lee, Jen-Chieh Lee, Pei-Jane Tsai, Wen-Chien Ko

**Affiliations:** 1Department of Internal Medicine, Tainan Hospital, Ministry of Health and Welfare, Tainan 700, Taiwan; yuebin16@yahoo.com.tw; 2Department of Internal Medicine, College of Medicine, National Cheng Kung University Hospital, National Cheng Kung University, Tainan 704, Taiwan; chichingbm85@yahoo.com.tw (C.-C.L.); jclee.eric@msa.hinet.net (J.-C.L.); 3Clinical Medicine Research Center, College of Medicine, National Cheng Kung University Hospital, National Cheng Kung University, Tainan 705, Taiwan; 4Graduate Institute of Medical Sciences, College of Health Sciences, Chang Jung Christian University, Tainan 711, Taiwan; 5Department of Medical Laboratory Science and Biotechnology, College of Medicine, National Cheng Kung University, Tainan 705, Taiwan; peijtsai@mail.ncku.edu.tw; 6Institute of Basic Medical Sciences, College of Medicine, National Cheng Kung University, Tainan 705, Taiwan; 7Department of Pathology, National Cheng Kung University Hospital, Tainan 705, Taiwan; 8Department of Medicine, College of Medicine, National Cheng Kung University, Tainan 705, Taiwan

**Keywords:** SARS-CoV-2, COVID-19, gut microbiome, probiotics, *Lactobacillus*, *Bifidobacteria*

## Abstract

Severe acute respiratory syndrome coronavirus 2 (SARS-CoV-2), an RNA virus of the family *Coronaviridae*, causes coronavirus disease 2019 (COVID-19), an influenza-like disease that chiefly infects the lungs through respiratory transmission. The spike protein of SARS-CoV-2, a transmembrane protein in its outer portion, targets angiotensin-converting enzyme 2 (ACE2) as the binding receptor for the cell entry. As ACE2 is highly expressed in the gut and pulmonary tissues, SARS-CoV-2 infections frequently result in gastrointestinal inflammation, with presentations ordinarily ranging from intestinal cramps to complications with intestinal perforations. However, the evidence detailing successful therapy for gastrointestinal involvement in COVID-19 patients is currently limited. A significant change in fecal microbiomes, namely dysbiosis, was characterized by the enrichment of opportunistic pathogens and the depletion of beneficial commensals and their crucial association to COVID-19 severity has been evidenced. Oral probiotics had been evidenced to improve gut health in achieving homeostasis by exhibiting their antiviral effects via the gut–lung axis. Although numerous commercial probiotics have been effective against coronavirus, their efficacies in treating COVID-19 patients remain debated. In ClinicalTrials.gov, 19 clinical trials regarding the dietary supplement of probiotics, in terms of *Lactobacillus* and mixtures of *Bifidobacteria* and *Lactobacillus*, for treating COVID-19 cases are ongoing. Accordingly, the preventive or therapeutic role of probiotics for COVID-19 patients can be elucidated in the near future.

## 1. Introduction

Severe acute respiratory syndrome coronavirus 2 (SARS-CoV-2), a new RNA virus of the family *Coronaviridae*, can cause coronavirus disease 2019 (COVID-19), majorly affecting pulmonary tissues by respiratory transmission [1,2]. Clinical presentations of COVID-19 vary greatly, ranging from no or mild symptoms often in young patients without comorbidities, moderate diseases with pneumonia, to severe diseases complicated by hypoxia, respiratory or multi-organ failure, and even death [2]. SARS-CoV-2 is composed of four structure proteins, including spike glycoproteins (S), small envelope glycoproteins (E), glycoproteins membrane (M), nucleocapsid (N), and other accessory proteins [3]. The spike protein of SARS-CoV-2, a transmembrane protein, uses angiotensin-converting enzyme 2 (ACE2) as the receptor of the cell entry [3,4]. In addition to extensive existence in pulmonary tissue, ACE2 is highly expressed in the gut [3,4]; therefore, in the human small intestinal organoids model, enterocytes are easily infected by SARS-CoV-2, as demonstrated by confocal and electron microscopy [1,5]. In the gut, ACE2 is not only a key regulator of dietary amino acid homeostasis, innate immunity, gut microbial ecology, and transmissible susceptibility to colitis [6], but also is linked to the activation of intestinal inflammation [6]. Accordingly, SARS-CoV-2 infections frequently result in gastrointestinal inflammation, with clinical presentations ranging from intestinal cramps and diarrhea to intestinal perforations (Figure 1) [7,8]. Additionally, its abdominal presentation was more frequent in critically ill patients requiring intensive care than those who did not require intensive care, and 10% of patients presented with diarrhea and nausea within 1–2 days before the development of fever and respiratory symptoms [9]. However, the evidence detailing successful therapy for gastrointestinal involvement in COVID-19 patients is currently limited.

One possible mechanism linked to gut presentations in COVID-19 is the downregulation of ACE2, followed by the decreased activation of mechanistic targets of rapamycin and increased autophagy, further leading to dysbiosis [7]. Another theory is that the blockage of ACE2 induces the increased levels of angiotensinogen by the hyperactivation of the renin–angiotensin system, resulting in the shutdown of the amino acid transporter BA0T1 and a lack of cellular tryptophan. These alterations cause the decreased secretion of antimicrobial peptides and disturbance in the gut microbiome [10]. Therefore, COVID-19 impacts the human gut microbiome, with a decline in microbial diversity and beneficial microbes [11].

## 2. The Interaction between Respiratory Tract Diseases and Gut Microbiota

A crucial association between a modified gut microbiome and the immune response to respiratory viral infections is evidenced. Taking respiratory syncytial virus and influenza as examples, gut microbiota was significantly altered by viral infections itself and multifactorial variables, such as inflammation-induced tumor necrosis factor-alpha (TNF-α) [12]. Intact microbiota provides signals leading to inflammasome activation, expression of pro-interleukin (IL)-1β and pro-IL-18, and the migration of dendritic cells (DCs) from the lung to the draining lymph node and T-cells, which are critical for protective immunity following influenza virus infection [13]. Disturbed gut microbiota directly or indirectly affects innate and adaptive immune signals and cells in the pulmonary tissue, such as the increased susceptibility to asthma, pulmonary allergic diseases, and chronic obstructive pulmonary diseases [14,15,16,17]. More importantly, the severity of influenza infections has been vastly related to the heterogeneous responses of the gut microbiota, as noted by the finding that *Bifidobacterium* species in the gut can expand to enhance host resistance to influenza [18].

In addition, gut microorganisms regulate innate memory by eliciting pattern recognition receptors (PRRs) on monocytes/macrophages and natural killer cells to recognize microbe- or pathogen-associated molecular patterns on microbes [19]. Toll-like receptors (TLRs) and nucleotide-binding oligomerization domain (NOD)-like receptors, recognizable on the host’s cells through PRRs, evoke different immunological reactions depending on the types of cells, ligands, or receptors [20]. The fine alteration of the regulatory balance of pro-inflammatory responses and inflammatory regulatory T cells (Tregs) ultimately controlled by the commensal microorganisms is critical in coordinating gut immune homeostasis [20,21]. For example, polysaccharide A, an immunomodulatory molecule, secreted by *Bacteroides fragilis,* can mediate the conversion of CD4+ T cells into IL-10-producing Foxp3(+) Treg cells, and may be considered for the prevention and treatment of experimental colitis in mice [21].

## 3. Gut Dysbiosis during COVID-19

Patients with COVID-19 had significant changes in fecal microbiomes, characterized by the enrichment of opportunistic pathogens and the depletion of beneficial commensals [22]. Dysbiosis has been vastly associated with COVID-19 severity [22,23,24,25], because the microbial diversity is regarded as a critical determinant of microbial ecosystem stability [26]. Among short-chain fatty acids (SCFAs), butyrate is not only responsible for energy requirements of the colonic epithelium, but also preserves tissues by mitigating chronic inflammatory responses through the regulation of pro- and anti-inflammatory cytokines [27]. Accordingly, decreases in the abundance of butyrate-producing bacteria (such as *Faecalibacterium prausnitzii* and *Clostridium* species), and the subsequent decline in SCFA availability have been correlated with severe COVID-19 [22,23,24,25,28,29]. Additionally, an increase in common pathogens in gut microbiota, such as *Prevotella*, *Enterococcus*, Enterobacteriaceae, or *Campylobacter*, were consistently associated with high infectivity, disease deterioration, or poor prognosis in COVID-19 patients [23,24,25,28]. The *Prevotella* species, for example, is associated with augmented T helper type 17 (Th17)-mediated mucosal inflammation, including activating TLR2 and Th17-polarizing cytokine production (such as IL-23 and IL-1), stimulating epithelial cells to produce IL-8, IL-6, and CCL20, and thus promoting neutrophil recruitment and inflammation [30]. The deterioration of the clinical course of patients with COVID-19 infection might be in part due to the activation of severe inflammation through disruption in gut microbiota and the out-growth of pathogenic bacteria.

Patients with COVID-19 also had the increased proportion of opportunistic fungal pathogens, such as *Aspergillus flavus* and *Aspergillus niger,* detected in fecal samples [31]. In metagenomic sequencing analyses of fecal samples from COVID-19 patients, the baseline abundance of *Coprobacillus*, *Clostridium ramosum*, and *Clostridium hathewayi* was correlated with disease severity, and an inverse correlation between abundance of *F. prausnitzii* (an anti-inflammatory bacterium) and disease severity was disclosed [22]. Furthermore, *Bacteroides dorei*, *Bacteroides thetaiotaomicron*, *Bacteroides massiliensis*, and *Bacteroides ovatus*, which downregulated the expression of ACE2 in the gut, were correlated inversely with SARS-CoV-2 load [22]. The same study team also indicated that, in the cases of active SARS-CoV-2 infections, the gut microbiota presented a higher abundance of opportunistic pathogens, while increased nucleotide and amino acid biosynthesis, as well as carbohydrate metabolism, were evidenced [24]. In summary, these findings reasonably suggest that the development of therapeutic agents able to neutralize the SARS-CoV-2 activity in the gut, as well as to restore the physiological gut microbiota composition, may be warranted.

A crucial association between the predominance of opportunistic pathogens in gut microbiomes and unfavorable outcomes of COVID-19 patients has been comprehensively reported [23]. In a Chinese cohort of COVID-19 patients with different disease severity, the abundance of butyrate-producing bacteria decreased significantly, which may help discriminate critically ill patients from general and severe patients. The increased proportion of opportunistic pathogens, such as *Enterococcus* and Enterobacteriaceae, in critically ill patients might be associated with a poor prognosis [23]. In another study, a higher abundance of opportunistic pathogens, such as *Streptococcus, Rothia, Veillonella,* and *Actinomyces* species, and a lower abundance of beneficial symbionts, could be noted in the gut microbiota of COVID-19 patients [25]. In the American cohort, the specific alteration in the gut microbiome, particularly *Peptoniphilus, Corynebacterium,* and *Campylobacter,* was also noticed [28]. Nevertheless, opportunistic pathogens were prevalent in the COVID-19 cases, particularly among critically individuals, but the causal effect of the predominance of opportunistic pathogens, and a grave outcome remains to be determined.

The recovery of dysbiosis after active SARS-CoV-2 infections exhibited geographical and demographic differences [22,28,32]. After the clearance of SARS-CoV-2 and resolution of respiratory symptoms, depleted symbionts and gut dysbiosis were usually persistent among recovered COVID-19 patients, because microbiota richness did not yield to normal levels after 6-month recovery [22]. In contrast, in an American cohort including recovered COVID-19 cases, the dysbiosis could rapidly recover with a return of the human gut microbiota to an uninfected status [28]. Although the great diversity in the ability of the microbiota return was disclosed, it was evident that the recovery of gut microbiota could be regarded as an indicator of the favorable prognosis among patients with COVID-19.

## 4. Therapeutic Effects of Dietary Supplement of Probiotics for COVID-19

Oral probiotics had been proven to exhibit antiviral effects and thereby to improve gut health for achieving homeostasis [33,34]. To take the influenza infection as an example, *Lactococcus lactis* JCM 5805 demonstrated the activity against influenza virus through the activation of anti-viral immunity [34]. The oral administration of *Bacteroides breve* YIT4064 can enhance antigen-specific IgG against influenza virus [33]. Moreover, a meta-analysis report indicated the administration of these probiotics significantly reduced the incidence of ventilator-associated pneumonia, possibly through reducing the overgrowth of potentially opportunistic pathogens and stimulating immune responses [35]. However, such a promotion of oral probiotics in treating critically ill patients experiencing COVID-19 should be further explored.

In COVID-19 patients, the excessive production of pro-inflammatory cytokines, a so-called “cytokine storm”, is pathologically related to acute respiratory distress syndrome and extensive tissue injury, multi-organ failure, or eventually death [36]. With COVID-19 progression, critically ill patients had higher plasma levels of many cytokines, in terms of IL-2, IL-7, IL-10, granulocyte colony-stimulating factor, IFN-γ-inducible protein-10, monocyte chemoattractant protein-1, macrophage inflammatory protein-1A, and TNF-α [37]. Therefore, therapeutic targeting on cytokines in COVID-19 treatment was evidenced to increase survival [36]. Fecal levels of IL-8 and IL-23 and intestinal specific IgA responses were vastly associated with severe COVID-19 disease, which indicated the co-existence of systemic and local intestine inflammation in critically ill patients [38]. One of the commercial probiotics, *Lactobacillus rhamnosus* HDB1258, might be effective in treating COVID-19 by modulating both microbiota-mediated immunity in gut and systemic inflammation induced by lipopolysaccharide [39]. Accordingly, concomitant targeting on local and systemic inflammatory responses by probiotics is reasonably believed to be valuable to counteract COVID-19-related gut and systemic inflammation.

Numerous probiotics and by-probiotic products exhibiting direct and indirect antiviral effects have been reported in the scientific literature. Lactic acid-producing bacteria such as *lactobacilli* can exert their antiviral activity by direct probiotic–virus interaction, the production of antiviral inhibitory metabolites, preventing secondary infection, and eliciting anti-viral immunity [40,41,42,43,44,45,46,47]. Nisin, one of the well-characterized bacteriocins from probiotics, contributes to probiotic antiviral effects against influenza A virus and other respiratory viruses [41,43]. A peptide, P18, produced by the probiotic *Bacillus subtilis* strain, was regarded as an antiviral compound against influenza virus [42]. Probiotics capsules containing live *B. subtilis* and *E. faecalis* (Medilac-S) can lower the acquisition of the gut colonization of potentially pathogenic microorganisms [44]. *L. rhamnosus* GG have been reported to prevent ventilator-associated pneumonia [45]. The heat-killed *L. casei* DK128 strain has been active against different subtypes of influenza viruses by an increasing proportion of alveolar macrophages in lungs and airways, the early induction of virus-specific antibodies, and reduced levels of pro-inflammatory cytokines and innate immune cells [46]. *S. salivarius* 24SMB and *S. oralis* 89a were able to inhibit the biofilm formation capacity of airway bacterial pathogens and even to disperse their pre-formed biofilms [47]. The *S. salivarius* strain K12 may stimulate IFN-γ release and suppress bronchial inflammation, and its colonization in the oral cavity and upper respiratory tract will actively interfere with the growth of pathogenic microbes [48]. Although these probiotics and their products provide the favorable antiviral interaction with immune composition in the gut, the feasibility and health effect of dietary probiotics to improve the dysbiosis in COVID-19 patients remains to be studied.

Numerous probiotics had been proposed to be beneficial in coronaviral infections, but the evidence detailing their efficacies in treating COVID-19 infection is limited [49]. *L. plantarum* Probio-38 and *L. salivarius* Probio-37 could inhibit transmissible gastroenteritis coronavirus [50]. The probiotic, *E. faecium* NCIMB 10415, has been approved as a feed additive for young piglets in the European Union for treating the transmissible coronavirus gastroenteritis [51]. The recombinant IFN-λ3-anchored *L. plantarum* can in vitro inhibit porcine gastroenteritis caused by coronavirus [52]. However, the clinical utility of probiotics in human infections caused by SARS-CoV-2 warrants further evaluations [53,54,55,56,57].

Another important issue regarding probiotics for COVID-19 cases is the patient safety. For an example, *B. longum* bacteremia had been reported in preterm infants receiving probiotics [58,59]. Since gastrointestinal SARS-CoV-2 involvement has been reported, the possibility of increased intestinal permeability should be expected and the risk of secondary bacterial infections in the gut is substantial if high-dosage steroid and other immunomodulation agents are administrated to treat the cytokine storm associated with COVID-19 [60,61]. The oral formulation Sivomixx®, which was a mixture of probiotics, was independently associated with a reduced risk for death in a retrospective, observational cohort study that included 200 adults with severe COVID-19 pneumonia [62]. In another study, nearly all COVID-19 patients treated with Sivomixx® showed remission of diarrhea and other symptoms within 72 h, in contrast to less than half in the control group [63]. However, the clinical application of probiotics in COVID-19 patients requires more evidence.

In ClinicalTrials.gov, 22 trials of probiotics for the prevention or adjuvant therapy of COVID-19 were registered since April 2020, including one aiming to study the effect of oxygen-ozone therapy, one studying intranasal probiotics, and the other using throat spray-containing probiotic [64]. Of the remaining 19 trials, 8 common probiotic strains include *Lactobacillus* (7 trials), a mixture of *Bifidobacteria* and *Lactobacillus* (5), and *Saccharomyces* species (2) (Table 1). The major outcome was greatly diverse in these trials, including disease prevention, symptom relief, antibody titers, disease progression, changes of viral load, microbiome effects, and mortality. Based on these trials, the role of dietary supplement probiotics for COVID-19 can be more evident in the near future.

There are microbiome-targeting agents other than oral probiotics for patients with COVID-19 infection. A clinical trial of oral prebiotics, KB109, a novel synthetic glycan to modulate gut microbiome composition and to increase SCFA production in the gut, is ongoing (NCT04414124) [64]. Throat spray containing three *Lactobacillus* strains was implemented in a clinical trial to change the severity of COVID-19 and prevent transmission of SARS-COV-2 virus to household members (NCT04793997) [64]. Moreover, there are several next-generation probiotics identified by metagenomic approaches, such as *F. prausnitzii* and *Akkermansia muciniphila*, which can generate diffusible metabolites, including butyrate, desaminotyrosine, and SCFAs, and may improve pulmonary immunity and prevent viral respiratory infections [65]. It can be expected, in the future, microbiome-targeting therapy may decrease disease severity, relief symptoms, or prevent viral transmission, and play a role in the treatment of patients with COVID-19 infection

## 5. Conclusions

Patients with COVID-19 had significant changes in fecal microbiomes, characterized by the enrichment of opportunistic pathogens and the depletion of beneficial commensals, which is vastly associated with disease severity. Besides anti-viral agents or supportive treatment, microbiome-targeting therapy may provide an alternative to prevent COVID-19 deterioration. Oral probiotics may have antiviral effects via the gut–lung axis and improve gut health for achieving homeostasis. Although some commercial probiotics have been effective against coronavirus, the evidence detailing their efficacies in treating COVID-19 patients is limited. Registered clinical trials of probiotics in COVID-19, mainly *Lactobacillus* and mixtures of *Bifidobacteria* and *Lactobacillus*, are ongoing and thus the preventive or therapeutic role of probiotics for such patients can be elucidated in the near future.

## Figures and Tables

**Figure 1 microorganisms-09-01605-f001:**
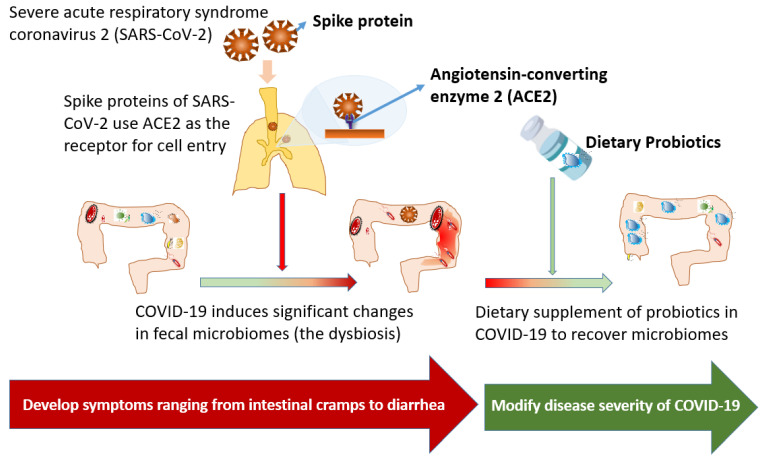
Gastrointestinal involvement and disturbance of gut microbiota during COVID-19 and recovery by dietary supplement of probiotics.

**Table 1 microorganisms-09-01605-t001:** Nineteen clinical trials of dietary supplement of probiotics in coronavirus disease 2019 (COVID-19) registered at *ClinicalTrials.gov* posted from April 2020 to June 2021.

ClinicalTrials.gov Identifier	Study Title	First Posted	Study Design	Probiotic Strain	Location	Outcome Measures	Status
NCT04366180	Evaluation of probiotic *Lactobacillus coryniformis* K8 on COVID-19 prevention in healthcare workers	28 April 2020	Randomized	*L. coryniformis* K8	Granada, Spain	Incidence of COVID-19 infection in healthcare workers	Recruiting
NCT04390477	Study to evaluate the effect of a probiotic in COVID-19	15 May 2020	Randomized	Not revealed	Alicante, Spain	ICU admission rate	Recruiting
NCT04399252	Effect of *Lactobacillus* on the microbiome of household contacts exposed to COVID-19	22 May 2020	Randomized	*L. rhamnosus* GG	North Carolina, United States	Incidence of symptoms of COVID-19	Active, not recruiting
NCT04420676	Synbiotic therapy of gastrointestinal symptoms during COVID-19 infection (SynCov)	9 June 2020	Randomized	Omni-Biotic® 10 AAD (chiefly *Lactobacillus* and *Bifidobacterium*)	Graz, Austria	Stool calprotectin	Recruiting
NCT04462627	Reduction of COVID 19 transmission to health care professionals	8 July 2020	Non-randomized	Metagenics Probactiol plus (chiefly *Lactobacillus* and *Bifidobacterium*)	Brussels, Belgium	Antibody concentration	Recruiting
NCT04507867	Effect of a NSS to reduce complications in patients with COVID-19 and comorbidities in stage III	11 August 2020	Randomized	*Saccharomyces bourllardii* with nutritional support system (NSS)	Mexico	Oxygen saturation	Not yet recruiting
NCT04517422	Efficacy of *L. plantarum* and *P. acidilactici* in adults with SARS-CoV-2 and COVID-19	18 August 2020	RCT	*L. plantarum* and *P. acidilactici*	Mexico City, Mexico	Severity progression of COVID-19	Completed
NCT04621071	Efficacy of probiotics in reducing duration and symptoms of COVID-19 (PROVID-19)	9 November 2020	RCT	Not revealed	Canada, Quebec	Duration of symptoms of the COVID-19	Recruiting
NCT04666116	Changes in viral load in COVID-19 after probiotics	14 December 2020	Randomized, single blind	GASTEEL PLUS (mixture of *Bifidobacteria* and *Lactobacillus*)	Valencia, Spain	Viral load in nasopharyngeal smear	Recruiting
NCT04734886	The effect of probiotic supplementation on SARS-CoV-2 antibody response after COVID-19	2 February 2021	Randomized	*L. reuteri* DSM 17938 + vitamin D	Örebro Län, Sweden	SARS-CoV-2 specific antibodies	Recruiting
NCT04756466	Effect of the consumption of a *Lactobacillus* strain on the incidence of COVID-19 in the elderly	16 February 2021	RCT	*Lactobacillus* strain	A Coruña, Spain	Incidence of SARS CoV-2 infection	Active, not recruiting
NCT04798677	Efficacy and tolerability of ABBC1 in volunteers receiving the influenza or COVID-19 Vaccine	15 March 2021	Non-randomized	*S. cerevisiae*, rich in selenium and zinc	Barcelona, Spain	Change in acute immune response to influenza vaccine after supplementation	Recruiting
NCT04813718	Post COVID-19 syndrome and the gut-lung axis	24 March 2021	Randomized	Omni-Biotic Pro Vi 5 (chiefly Lactobacillus)	Graz, Austria	Microbiome composition	Recruiting
NCT04847349	Live microbials to boost anti-severe acute respiratory syndrome coronavirus-2 (SARS-CoV-2) immunity clinical trial	19 April 2021	RCT	OL-1 (Content not revealed)	New Jersey, United States	Change in serum titer of anti-SARS-CoV-2 IgG	Recruiting
NCT04854941	Efficacy of probiotics in the treatment of hospitalized patients with novel coronavirus infection	22 April 2021	Randomized	*L. rhamnosus, B. bifidum, B. longum* subsp. *infantis* and *B. longum*	Moscow, Russian	Mortality	Completed
NCT04877704	Symprove (Probiotic) as an add-on to COVID-19 management	7 May 2021	Randomized	Symprove ( *L. rhamnosus, E. faecium*, *L. acidophilus* and *L. plantarum*)	London, United Kingdom	Length of hospital stay	Not yet recruiting
NCT04884776	Modulation of gut microbiota to enhance health and immunity	13 May 2021	RCT	Probiotics blend (3 *Bifidobacteria*)	Hong Kong	Restoration of gut dysbiosis	Not yet recruiting
NCT04907877	*Bifidobacteria* and *Lactobacillus* in symptomatic adult COVID-19 outpatients (ProCOVID)	1 June 2021	Randomized	NordBiotic ImmunoVir (mixture of *Bifidobacteria* and *Lactobacillus*)	Not revealed	Global symptom score	Not yet recruiting
NCT04922918	*Ligilactobacillus salivarius* MP101 for elderly in a nursing home (PROBELDERLY)	11 June 2021	Single group	*Ligilactobacillus salivarius* MP101	Madrid, Spain	Barthel index, functional status score	Recruiting

RCT: randomized controlled trial; ICU: intensive care unit; IgG: immunoglobulin G.

## Data Availability

Data available in a publicly accessible repository.

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
