# Peer review of "Gut Dysbiosis during COVID-19 and Potential Effect of Probiotics"

_microorganisms, 2021, doi:10.3390/microorganisms9081605_

Round 1

Reviewer 1 Report

The review by Yuan-Pin Hung et al. reports the role of oral probiotics in severe acute respiratory syndrome coronavirus 2.

The authors well explain the immunological and pathogenic mechanisms of the disease and microbiomes’ change (dysbiosis) in the gut-lung axis. The authors presents the results of 22 trials of probiotics for the prevention or adjuvant therapy of COVID-19. The most common probiotic regimens include Lactobacillus, the association between Lacotobacillus and Bifidobacteria, and Saccaromyces species.

The role of these probiotics for COVID-19 remains unclear and it can be elucidated in the next future.

The paper is well designed and written. It can be accepted in the present form.

Author Response

Dear the editor and reviewer of Microorganisms,

Enclosed, please find our revised manuscript entitled "Gut dysbiosis during COVID-19 and potential effect of probiotics ". We have considered very carefully for the concerns raised by the reviewers and made responsive alterations in the revised manuscript. We hope that our changes have satisfactorily clarified the points raised.

In answering the reviewers’ concerns, we made point-by-point responses to the comments summarized below. The revisions made in the manuscript were marked in red color. We appreciate the reviewers and the editor for their careful reading and insightful comments, and make our best efforts to improve the manuscript by responding to their comments.

We look forward to hearing from you.

Best Regards,

Wen-Chien Ko, MD

Division of Infectious Diseases, Department of Internal Medicine, National Cheng Kung University Hospital, No. 138, Sheng Li Road, Tainan, 70403, Taiwan

TEL: 886-6-2353535, ext. 3596   FAX: 886-6-2752038

E-mail: winston3415@gmail.com

Reviewer 1:

The review by Yuan-Pin Hung et al. reports the role of oral probiotics in severe acute respiratory syndrome coronavirus 2.

The authors well explain the immunological and pathogenic mechanisms of the disease and microbiomes’ change (dysbiosis) in the gut-lung axis. The authors presents the results of 22 trials of probiotics for the prevention or adjuvant therapy of COVID-19. The most common probiotic regimens include Lactobacillus, the association between Lacotobacillus and Bifidobacteria, and Saccaromyces species.

The role of these probiotics for COVID-19 remains unclear and it can be elucidated in the next future.

The paper is well designed and written. It can be accepted in the present form.

Reply: We appreciate the reviewer for the positive feedback.

Reviewer 2 Report

I appreciate all the hard work the study group invested in this review. The topic is obviously hot enough and the use of probiotics in this setting has raised very frequent questions from both physicians and people across the globe (not only already infected, but also in good health, trying to use probiotics in order to prevent the infection with SARS-CoV-2): to use or not to use probiotics? If yes, how do they help? If yes, which strain(s) of probiotics to use and what dose and for how long? Unfortunately, the answers to the questions are not found in this review, as many studies are still ongoing, but I appreciate that these studies are presented in detail in Table 1. Generally, this manuscript is correct, written in an active style, with attention to details and it appears clear. Even though there are many data, the manuscript appears easily to be followed. Very nicely structured and organized. Introduction: well written, with appropriate recent references.  The interaction between respiratory tract diseases and gut microbiota:  I appreciate the details regarding the impact of respiratory viral infections on gut microbiota. Dysbiosis during COVID-19: paragraph with excellent references, nicely conceived and presented, including associations with severity of the disease and presenting also the time to recovery of gut dysbiosis after COVID-19. Therapeutic effects of dietary supplement of probiotics for COVID-19: properly introducing studies in vitro, in animals and in humans, however many other references could have been included.

Minor comments:

  1. Title: I suggest adding “Gut” or “Intestinal” before dysbiosis, as the paper focuses on this.
  2. Keywords: I suggest adding “gut” before microbiome or “intestinal”.
  3. Introduction: line 65- please insert 1, after Figure.
  4. Dysbiosis during COVID-19: line 177: please replace “was” with ‘were”.
  5. Although the authors used updated reference, well chosen, to support their reviews, I would suggest the authors to take a look at the following manuscripts, already published and maybe insert some of them. There are also good reviews on probiotic use and in my opinion the authors should acknowledge these previous reviews.

*Xu K, Cai H, Shen Y, et al. Management of corona virus disease-19 (COVID-19): the Zhejiang experience. J. Zhejiang Univ. (medical science 2020; 49.

* Feng Z, Wang Y, Qi W. The small intestine, an underestimated site of SARS-CoV-2 infection: from red queen effect to probiotics. DOI:10.20944/preprints202003.0161.v1 (well, this is only in preprint form)

* Mak JWY, Chan FKL, Ng SC. Probiotics and COVID-19: one size does not fit all. Lancet Gastroenterol Hepatol. 2020 Jul;5(7):644-645.

* Bottari B, et al. Probiotics and Covid-19, International Journal of Food Sciences and Nutrition 2021; 72:3, 293-299

*Kurian et al. Probiotics in Prevention and Treatment of COVID-19: Current Perspective and Future Prospects. Archives of Medical Research 2021;16:31. (VERY GOOD ONE!, including figures)

*Gou W, et al. Gut microbiota may underlie the predisposition of healthy individuals to COVID-19. medRxiv2020

*Peng J, et al. Probiotics as Adjunctive Treatment for Patients Contracted COVID-19: Current Understanding and Future Needs. Front Nutr 2021

*Sundararaman A, et al. Role of probiotics to combat viral infections with emphasis on COVID-19. Appl. Microbiol. Biotechnol 2020; 8089-8104.

*Khaled JMA. Probiotics, prebiotics, and COVID-19 infection: A review article. Saudi Journal of Biological Sciences 2021; 28 (1): 865-869.

Author Response

Dear the editor and reviewer of Microorganisms,

Enclosed, please find our revised manuscript entitled "Gut dysbiosis during COVID-19 and potential effect of probiotics ". We have considered very carefully for the concerns raised by the reviewers and made responsive alterations in the revised manuscript. We hope that our changes have satisfactorily clarified the points raised.

In answering the reviewers’ concerns, we made point-by-point responses to the comments summarized below. The revisions made in the manuscript were marked in red color. We appreciate the reviewers and the editor for their careful reading and insightful comments, and make our best efforts to improve the manuscript by responding to their comments.

We look forward to hearing from you.

Best Regards,

Wen-Chien Ko, MD

Division of Infectious Diseases, Department of Internal Medicine, National Cheng Kung University Hospital, No. 138, Sheng Li Road, Tainan, 70403, Taiwan

TEL: 886-6-2353535, ext. 3596   FAX: 886-6-2752038

E-mail: winston3415@gmail.com

Reviewer 2:

I appreciate all the hard work the study group invested in this review. The topic is obviously hot enough and the use of probiotics in this setting has raised very frequent questions from both physicians and people across the globe (not only already infected, but also in good health, trying to use probiotics in order to prevent the infection with SARS-CoV-2): to use or not to use probiotics? If yes, how do they help? If yes, which strain(s) of probiotics to use and what dose and for how long? Unfortunately, the answers to the questions are not found in this review, as many studies are still ongoing, but I appreciate that these studies are presented in detail in Table 1. Generally, this manuscript is correct, written in an active style, with attention to details and it appears clear. Even though there are many data, the manuscript appears easily to be followed. Very nicely structured and organized. Introduction: well written, with appropriate recent references.  The interaction between respiratory tract diseases and gut microbiota:  I appreciate the details regarding the impact of respiratory viral infections on gut microbiota. Dysbiosis during COVID-19: paragraph with excellent references, nicely conceived and presented, including associations with severity of the disease and presenting also the time to recovery of gut dysbiosis after COVID-19. Therapeutic effects of dietary supplement of probiotics for COVID-19: properly introducing studies in vitro, in animals and in humans, however many other references could have been included.

Minor comments:

  1. Title: I suggest adding “Gut” or “Intestinal” before dysbiosis, as the paper focuses on this.

Reply: Thanks for the valuable suggestion. Gut” was added before dysbiosis in the title, running title, and line 115.

  1. Keywords: I suggest adding “gut” before microbiome or “intestinal”.

Reply: “Gut” was added before microbiome, as suggested in line 48.

  1. Introduction: line 65- please insert 1, after Figure.

Reply: The figure number was added in line 69.

  1. Dysbiosis during COVID-19: line 177: please replace “was” with ‘were”.

Reply: The grammar is revised as suggested in line 170.

  1. Although the authors used updated reference, well chosen, to support their reviews, I would suggest the authors to take a look at the following manuscripts, already published and maybe insert some of them. There are also good reviews on probiotic use and in my opinion the authors should acknowledge these previous reviews.

Reply: These good reviews on probiotic use were cited in this article (Ref. 29, 49, 53-57).

Reviewer 3 Report

Line 65: please replace “Figure” with “Figure 1”

Lines 102-105: Too long phrase. Please rephrase this sentence in order to make it more intelligible.

Lines 112-115: Too long phrase. Please rephrase this sentence in order to make it more intelligible

Line 119: The term dysbiosis has been just introduced in line 74. Please remove

Line 122: Please replace “in general” with "Among short-chain fatty acids (SCFA)"

Lines 125-128: Please rephrase as follows: Accordingly, decreases in the abundance of butyrate-producing bacteria (such as Faecalibacterium prausnitzii and Clostridium species), and the subsequent decline in the SCFAs availability have been correlated with severe COVID-19.

Lines 150-154: Please rephrase as follows: The same study team also indicated that, in the cases of active SARS-CoV-2 infections, the gut microbiota presented a higher abundance of opportunistic pathogens, while increased nucleotide and amino acid biosynthesis, as well as, carbohydrate metabolism were evidenced.

Lines 154-157: Please move this sentence at the end of the paragraph and rephrase it as follows: In sum, these findings reasonably suggest that the development of therapeutic agents able in neutralizing the SARS-CoV-2 activity in the gut, as well as, to restore the physiological gut microbiota composition may be warranted.

Line 192: pleased add “the” before “administration”

Line 192: please insert comma after “pneumonia”

Line 194: please replace "potential" with "potentially"

Lines 205-2011: Too long phrase. Please rephrase to make it more intelligible

Lines 215-216: Please rephrase as follows: Numerous probiotics and by-probiotic products exhibiting direct and indirect antiviral effects have been reported in the scientific literature.

Line 216: please rephrase as follows: Lactic acid-producing bacteria such as lactobacilli

Line 240: please add “be” before “beneficial”

Line 256: please replace “for the cytokine storm of COVID-19” with "to treat the cytokine storm associated with COVID-19"

Line 256: add “The” before “oral”

Line 267: please replace "probiotics" with "eight probiotic strains"

Line 259: An additional study evidencing the effectiveness of SIVOMIXX against SARS-CoV-2 infections has been published, doi: 10.3389/fmed.2020.00389. I suggest introducing such paper since it underlines the short period beneficial effects of probiotics on gastrointestinal symptoms associated with COVID-19.

Author Response

Dear the editor and reviewer of Microorganisms,

Enclosed, please find our revised manuscript entitled "Gut dysbiosis during COVID-19 and potential effect of probiotics ". We have considered very carefully for the concerns raised by the reviewers and made responsive alterations in the revised manuscript. We hope that our changes have satisfactorily clarified the points raised.

In answering the reviewers’ concerns, we made point-by-point responses to the comments summarized below. The revisions made in the manuscript were marked in red color. We appreciate the reviewers and the editor for their careful reading and insightful comments, and make our best efforts to improve the manuscript by responding to their comments.

We look forward to hearing from you.

Best Regards,

Wen-Chien Ko, MD

Division of Infectious Diseases, Department of Internal Medicine, National Cheng Kung University Hospital, No. 138, Sheng Li Road, Tainan, 70403, Taiwan

TEL: 886-6-2353535, ext. 3596   FAX: 886-6-2752038

E-mail: winston3415@gmail.com

Reviewer 3

Line 65: please replace “Figure” with “Figure 1”

Reply: The figure number was added in line 69.

Lines 102-105: Too long phrase. Please rephrase this sentence in order to make it more intelligible.

Reply: The sentence was revised as “In addition, gut microorganisms regulate innate memory by eliciting pattern recognition receptors (PRRs) on monocytes/macrophages and natural killer cells to recognize microbe- or pathogen-associated molecular patterns on microbes” (line 101-103).

Lines 112-115: Too long phrase. Please rephrase this sentence in order to make it more intelligible

Reply: It had been rephrased as “For example, polysaccharide A, an immunomodulatory molecule, secreted by Bacteroides fragilis, can mediate the conversion of CD4+ T cells into IL-10-producing Foxp3(+) Treg cells, and may be considered for the prevention and treatment of experimental colitis in mice” in line 110-113.

Line 119: The term dysbiosis has been just introduced in line 74. Please remove

Reply: It had been removed in line 116.

Line 122: Please replace “in general” with "Among short-chain fatty acids (SCFA)"

Reply: It had been replaced by “Among short-chain fatty acids (SCFA)” in line 120.

Lines 125-128: Please rephrase as follows: Accordingly, decreases in the abundance of butyrate-producing bacteria (such as Faecalibacterium prausnitzii and Clostridium species), and the subsequent decline in the SCFAs availability have been correlated with severe COVID-19.

Reply: It was revised as suggested inline 123-126.

Lines 150-154: Please rephrase as follows: The same study team also indicated that, in the cases of active SARS-CoV-2 infections, the gut microbiota presented a higher abundance of opportunistic pathogens, while increased nucleotide and amino acid biosynthesis, as well as, carbohydrate metabolism were evidenced.

Reply: It was revised as suggested in line 145-149.

Lines 154-157: Please move this sentence at the end of the paragraph and rephrase it as follows: In sum, these findings reasonably suggest that the development of therapeutic agents able in neutralizing the SARS-CoV-2 activity in the gut, as well as, to restore the physiological gut microbiota composition may be warranted.

Reply: It was revised as suggested in line 149-151.

Line 192: pleased add “the” before “administration”

Reply: We revise the sentence as suggested in line 184.

Line 192: please insert comma after “pneumonia”

Reply: A comma was added after “pneumonia” in line 184.

Line 194: please replace "potential" with "potentially"

Reply: The wording was revised as suggested in line 186.

Lines 205-2011: Too long phrase. Please rephrase to make it more intelligible

Reply: The sentence was revised as “Fecal levels of IL-8 and IL-23 and intestinal specific IgA responses were vastly associated with severe COVID-19 disease, which indicated the co-existence of systemic and local intestine inflammation in critically ill patients [38]. One of the commercial probiotics, Lactobacillus rhamnosus HDB1258, might be effective in treating COVID-19 by modulating both microbiota-mediated immunity in gut and systemic inflammation induced by lipopolysaccharide [39].” in line 196-201.

Lines 215-216: Please rephrase as follows: Numerous probiotics and by-probiotic products exhibiting direct and indirect antiviral effects have been reported in the scientific literature.

Reply: The sentence was revised as suggested in line 205-206.

Line 216: please rephrase as follows: Lactic acid-producing bacteria such as lactobacilli

Reply: It was revised as suggested in line 206-207.

Line 240: please add “be” before “beneficial”

Reply: It was revised as suggested in line 228.

Line 256: please replace “for the cytokine storm of COVID-19” with "to treat the cytokine storm associated with COVID-19"

Reply: It was revised as suggested in line 242-243.

Line 256: add “The” before “oral”

Reply: It was revised as suggested in line 243.

Line 267: please replace "probiotics" with "eight probiotic strains"

Reply: It was revised as “eight common probiotic strains” in line 254-255.

Line 259: An additional study evidencing the effectiveness of SIVOMIXX against SARS-CoV-2 infections has been published, doi: 10.3389/fmed.2020.00389. I suggest introducing such paper since it underlines the short period beneficial effects of probiotics on gastrointestinal symptoms associated with COVID-19.

Reply: The study evidencing the effectiveness of SIVOMIXX against SARS-CoV-2 infections was cited as Ref. 63 in line 246-248.

Reviewer 4 Report

The article describes the current state of knowledge on the relationship between Coivd 19 and gut microbiota. Due to the still present coronavirus threat, this topic is of interest to practically all people. It is clearly written and consists of 4 chapters. The introduction describes the way the virus enters the intestine. Then, the interaction between the covid-19 virus and the intestinal bacteria is explained, and studies are presented that show a negative effect on the intestinal microflora during and after coronavirus infection. Finally, studies are shown describing the beneficial effects of probiotic supplementation on the course of Covid-19. I suggest to cite a recent article DOI: 10.3390 / microorganisms9050941, which describes "Next-Generation Probiotics" in covid -19.

More detailed comments:
Line 37 - several - please replace with current number of studies
line 65 Figure 1
line 101-102- please replace "gut microbiota" with the synonym
line 107nucleotide-binding receptor (NOD) - please check if it is a correct nomenclature
line 160-164 - please modify the sentence to improve the style

Author Response

Dear the editor and reviewer of Microorganisms,

Enclosed, please find our revised manuscript entitled "Gut dysbiosis during COVID-19 and potential effect of probiotics ". We have considered very carefully for the concerns raised by the reviewers and made responsive alterations in the revised manuscript. We hope that our changes have satisfactorily clarified the points raised.

In answering the reviewers’ concerns, we made point-by-point responses to the comments summarized below. The revisions made in the manuscript were marked in red color. We appreciate the reviewers and the editor for their careful reading and insightful comments, and make our best efforts to improve the manuscript by responding to their comments.

We look forward to hearing from you.

Best Regards,

Wen-Chien Ko, MD

Division of Infectious Diseases, Department of Internal Medicine, National Cheng Kung University Hospital, No. 138, Sheng Li Road, Tainan, 70403, Taiwan

TEL: 886-6-2353535, ext. 3596   FAX: 886-6-2752038

E-mail: winston3415@gmail.com

Reviewer 4

The article describes the current state of knowledge on the relationship between Coivd 19 and gut microbiota. Due to the still present coronavirus threat, this topic is of interest to practically all people. It is clearly written and consists of 4 chapters. The introduction describes the way the virus enters the intestine. Then, the interaction between the covid-19 virus and the intestinal bacteria is explained, and studies are presented that show a negative effect on the intestinal microflora during and after coronavirus infection. Finally, studies are shown describing the beneficial effects of probiotic supplementation on the course of Covid-19. I suggest to cite a recent article DOI: 10.3390 / microorganisms9050941, which describes "Next-Generation Probiotics" in covid -19.

Reply: The article related to next-generation probiotics in COVID-19 had been cited as Ref. 65, and one sentence was added to explain the idea of next-generation probiotics in line 267-271.

More detailed comments:
Line 37 - several - please replace with current number of studies

Reply: It was replaced by the number of 19 in line 42.

line 65 Figure 1

Reply: It was revised as Figure 1 in line 69.

line 101-102- please replace "gut microbiota" with the synonym

Reply: It had been replaced with “gut microorganisms” in line 101.

line 107 nucleotide-binding receptor (NOD) - please check if it is a correct nomenclature
Reply: It was revised as “nucleotide-binding oligomerization domain (NOD)-like receptors” in line 104-105.

line 160-164 - please modify the sentence to improve the style

Reply: We revised it as “In a Chinese cohort of COVID-19 patients with different disease severity, the abundance of butyrate-producing bacteria decreased significantly, which may help discriminate critically ill patients from general and severe patients. The increased proportion of opportunistic pathogens, such as Enterococcus and Enterobacteriaceae, in critically ill patients might be associated with a poor prognosis [23]. In another study, a higher abundance of opportunistic pathogens, such as Streptococcus, Rothia, Veillonella, and Actinomyces species, and a less abundance of beneficial symbionts, could be noted in the gut microbiota of COVID-19 patients [25].” in line 154-161.